# Validity of the International Physical Activity Questionnaire Long Form for Assessing Physical Activity and Sedentary Behavior in Subjects with Chronic Stroke

**DOI:** 10.3390/ijerph18094729

**Published:** 2021-04-29

**Authors:** Maria-Arantzazu Ruescas-Nicolau, María Luz Sánchez-Sánchez, Sara Cortés-Amador, Sofía Pérez-Alenda, Anna Arnal-Gómez, Assumpta Climent-Toledo, Juan J. Carrasco

**Affiliations:** 1Physiotherapy in Motion, Multispeciality Research Group (PTinMOTION), Department of Physiotherapy, University of Valencia, 46010 Valencia, Spain; Arancha.Ruescas@uv.es (M.-A.R.-N.); Sofia.Perez-Alenda@uv.es (S.P.-A.); asclito@alumni.uv.es (A.C.-T.); Juan.J.Carrasco@uv.es (J.J.C.); 2Research Unit in Clinical Biomechanics-UBIC, Department of Physiotherapy, University of Valencia, 46010 Valencia, Spain; Sara.Cortes@uv.es (S.C.-A.); Anna.Arnal@uv.es (A.A.-G.); 3Intelligent Data Analysis Laboratory, University of Valencia, 46100 Burjassot, Spain

**Keywords:** stroke, physical activity, sedentary behavior, international physical activity questionnaire (IPAQ), validity, accelerometer, self-report

## Abstract

Validation studies of questionnaires used to assess physical activity (PA) and sedentary behavior (SB) in stroke survivors are scarce. This cross-sectional study aimed to examine the validity of the International Physical Activity Questionnaire long-form (IPAQ-LF) in community living adults with post-stroke sequelae (≥6 months) and preserved ambulation. Participants’ functional mobility, lower limb strength, ambulatory level, stroke severity, and disability were assessed. An accelerometer (ActiGraph GT3X+) was worn for ≥7 consecutive days. Subsequently, the IPAQ-LF was interview-administered. Fifty-six participants (58.1 ± 11.1 years, 66.1% male) were included. A strong correlation between the two methods was found for total PA time (ρ = 0.55, *p* < 0.001). According to the Bland-Altman analyses, over-reporting moderate-to-vigorous PA and under-reporting total PA in the IPAQ-LF were found in those participants with higher PA levels. Both methods measured sedentary time similarly, though random error was observed between them. Moderate-strong correlations were found between the IPAQ-LF and physical function (ρ = 0.29–0.60, *p* < 0.05). In conclusion, in people with chronic stroke, the IPAQ-LF presented acceptable levels of validity for estimating total PA time in those who are insufficiently active. Therefore, it could be a useful tool to screen for inactive individuals with chronic stroke who can benefit from PA interventions addressed to implement healthier lifestyles.

## 1. Introduction

At present, stroke is considered the second major cause of long-term disability worldwide [1]. After stroke, motor sequelae decrease survivors’ physical activity (PA) level, affecting their independence and quality of life [2,3]. In fact, post-stroke subjects in chronic phase attain less than half of the daily steps when compared with healthy counterparts [4,5], and only 17% of them achieved the international recommendations for PA [6]. Additionally, stroke survivors spend about 11 h/day in sedentary behavior (SB) (74.8% of waking hours) [7] and are 1.2 h/day more sedentary than healthy peers [6]. In this population, low PA levels and greater amounts of SB have shown to independently increase the risk of cardiovascular diseases [8,9] and the recurrence of stroke [10]. In this regard, adequate assessment methods are essential to precisely capture PA and sedentary habits in people post-stroke as they behave in their usual free-living environment, and to design interventions addressed to modify these practices.

To measure PA and SB, objective (direct) and subjective (indirect) methods can be used. Among the former, accelerometry has been used as a measurement for estimating time spent in PA and in SB [11]. A major advantage of accelerometers relies on their increased precision and accuracy for measuring usual free-living PA, since recall error and bias are less likely to occur [12]. Nonetheless, this measure hardly quantifies activities such as carrying heavy loads, weight lifting, or cycling. In addition, data derived from the vertical axis limits the ability of the accelerometer to notice movement of the upper body when configured around the pelvic girdle [13] and movement during walking in people with stroke since maximal range of motion occurs along the sagittal plane (medium-lateral axis) [14]. Likewise, a 7-day wear period is required to perform a single evaluation and the monitor handling and data processing need expertise [15], which decreases its feasibility in clinical settings.

On the other hand, indirect subjective measurements of PA and SB include the fulfillment of standardized self-report questionnaires, which are simple, inexpensive, efficient and easy to administer. However, they are influenced by the individuals’ cognitive function and their memory recall accuracy as well as social desirability [16], which has been shown to increase the likelihood of over- and underestimation of PA and sedentary habits in elders [12,17]. Consequently, there is growing need to assess the validity of self-reported measurements in different populations.

Among PA questionnaires, the International Physical Activity Questionnaire (IPAQ; www.ipaq.ki.se (accessed on 8 April 2021)) is one of the most commonly used in adults [18]. Designed to allow cross-national comparisons, this questionnaire has a short-form and a long-form, which gathers larger information on the PA context [19]. In chronic stroke, this detailed PA context would be highly relevant for approaching effective and context-specific interventions.

The validity of the IPAQ long-form (IPAQ-LF) has been assessed against accelerometer-derived measures in less physically active people, such as elders [15,17,20,21,22] and disease-related populations [23,24,25,26]. However, to the best of our knowledge, no previous research has analyzed the validity of the IPAQ-LF against objective measurements in people with chronic stroke. Therefore, this study aimed to examine the validity of the IPAQ-LF (interviewed-administered, last 7 days) compared to accelerometer-derived measures to assess PA and SB in subjects with chronic stroke. Based on the validity observed in the elderly population [27], it was hypothesized that the IPAQ-LF would show moderate/acceptable validity (correlation coefficient ≥ 0.41) in our population. In addition, since levels of PA and sedentary habits are affected by physical impairments after stroke [3,28], a second aim of this study was to assess validity by analyzing the association of the IPAQ-LF-derived PA and SB measures with physical function measures of chronic stroke survivors.

## 2. Materials and Methods

### 2.1. Study Design and Participants

A cross-sectional observational study was performed in which individuals with chronic stroke sequelae (onset ≥ 6 months), aged over 18 years, were recruited from several brain injury associations, physiotherapy clinics, and community facilities for people with neurological disability located in the Valencia region (Spain). Subjects were included if they lived at home for at least 2 months since their most recent stroke, could walk independently around the house with or without a mobility aid but without requiring supervision from another person (Functional Ambulation Classification of Hospital of Sagunto [29] ≥2), and had sufficient cognitive understanding to provide informed consent and comply with the assessment procedures. Fulfillment of the selection criteria was performed by a physical therapist with more than 15 years of clinical experience with the stroke population. Exclusions were made in the presence of any locomotor, cardiac, neurological, communication, or cognitive disorders other than stroke that prevented the individual from performing the study tests and questionnaires. The study took place from January to March and from July to November 2020. All study participants were fully informed about the study purpose and experimental procedures and provided written informed consent. The study was conducted in accordance with the Declaration of Helsinki [30] and the protocol was approved by the authors’ institution Ethics Committee (ID no. 1563377228465). This manuscript adheres to the Strengthening the Reporting of Observational Studies in Epidemiology (STROBE) guidelines [31].

### 2.2. Procedures

For the purposes of this study, two assessment visits were scheduled 8 days apart in the author’s University laboratory or at the community facility attended by participants (Figure 1). On the first visit, participants’ demographic and clinical characteristics were collected through medical records or by interview. Afterwards, weight and height were measured and cognitive function was assessed with the Montreal Cognitive Assessment (MoCA) [32]. Next, physical function tests were performed to assess: functional mobility with the timed up and go test (TUG) [33]; lower limb strength with the five times sit-to-stand test (5 × STS) [34]; ambulatory level with the 10-m walk test (10 MWT) [35]; stroke severity with the Stroke Impact Scale 16 (SIS-16) [36]; and disability with the Modified Rankin Scale (MRS) [37]. Before ending this session, participants were instructed to wear an accelerometer for at least 7 consecutive days and to keep an activity diary. On the second visit, participants returned the activity diary and accelerometer and subsequently completed the IPAQ-LF. This enabled comparative analysis between accelerometer data and IPAQ-LF data.

#### 2.2.1. Accelerometry

In this study, the ActiGraph GT3X+ accelerometers (ActiGraph Inc., Pensacola, FL, USA) were used to objectively assess participants’ PA and SB. They were set to record three axes of acceleration (e.g., vertical, anterior-posterior and medium-lateral) at 30 Hz (30 times per s) with data smoothed using 1 s epochs. For a free-living week (7 consecutive days), participants were instructed to wear the device on an elastic belt placed at the height of the iliac crest of the non-paretic side, except when sleeping at night, showering or doing any water-based activity. When receiving the accelerometer, participants were shown and carefully instructed, verbally and in writing, on how to properly wear and handle the monitor. Also, it was made clear that they had to keep their PA and SB as usual as possible during the study period.

Raw accelerometer data (activity counts) were downloaded via ActiLife software (v 6.13.4, ActiGraph, Pensacola, FL, USA) for processing. Days were excluded from the dataset if the monitor wear time was <240 min/day. Requirements for participants’ data to be included in the analyses implied 3 valid days out of the 7-day period (containing one weekend day), with ≥480 min of wear time each day [28,38,39], given that 3-day accelerometer data have proved valid and reliable when measuring free-living PA [40,41]. If there were ≥3 valid days available, since PA habits may be affected by wearing the device when starting data collection [42], the first day was excluded. The participants’ activity diaries were used to refine night sleep time during the monitoring period, thus custom-made filters were developed to determine daily wake hours. In addition, accelerometer non-wear time was defined as periods of ≥90 min of consecutive zeroes and were not included in the analysis [43]. In agreement with previous research work about actigraphy in stroke population and elders [28,40,44], Freedson et al.’s [45] cut-points were applied to the data to classify PA intensity as sedentary (≤99 counts/min), light (100–1951 counts/min) and moderate-to-vigorous (≥1952 counts/min) (MVPA). Participants were excluded from the analysis if they did not meet valid wear criteria (e.g., day and/or hours).

To match with IPAQ-LF outcome measures (see Section 2.2.2), accelerometer PA data was calculated as minutes per week with the equation (total PA(min)/no. of valid days) × 7 days [15] and was grouped by intensity, determining the following variables: total time spent in SB (min/day); total time spent in light-intensity PA (LPA) (min/week); and total time spent in MVPA (min/week). Also, accelerometer total PA time was computed as the time spent in LPA plus the time spent in MVPA (min/week). In addition, compliance with the PA recommendations for stroke survivors of 150 min per week in MVPA [46] was determined.

#### 2.2.2. Self-Reported Physical Activity

The Spanish version of the IPAQ-LF was administered by trained interviewers to assess participants’ self-report PA and SB. It is a 27-item questionnaire in which duration (hours and minutes per day), frequency (times per week), and intensity (walking, moderate and vigorous) of the previous 7 days’ PA within four different domains (job-related, transport-related, domestic and leisure-time PA) is collected. Activity should last a minimum of 10 consecutive minutes to be registered. Sedentary time (seated or reclined posture during waking hours) on a usual weekday and weekend day in the same time period is also recorded.

The IPAQ guidelines [47] were followed to depurate the data, with the exception of the truncation rule for high values (activities exceeding 180 min/day). As performed in other validation studies of the IPAQ in neurologic population [48], truncation was dismissed for two reasons. First, we considered that it would bias our participants’ real time spent in PA as reported in the IPAQ when compared to accelerometry data, which considers all of the PA time performed by the subject. Second, truncation is not usually used in clinical settings to assess PA. Then, to more reliably reflect the use of this questionnaire in clinical practice and to improve its applicability to this scope, truncation was not performed. Hence, high values were included if no sound reason for exclusion was found. Thus, within each intensity and across domains, data from the questionnaire were added to estimate the total amount of time (min/week) spent walking, in moderate-intensity PA (MPA, metabolic equivalent value 4–6 METs (an international unit of measuring metabolic energy)) and in vigorous-intensity PA (VPA, metabolic equivalent value > 6 METs). Time derived from the MPA and VPA domains were summed to estimate time spent in MVPA (metabolic equivalent value > 3 METs). In addition, total PA time (min/week) was estimated by adding up the time spent walking and in MVPA, as indicated by the scoring protocol [47]. Regarding SB, the weighted average of the self-reported time spent sitting or lying during waking hours on a usual weekday and weekend day was calculated (min/day), including time during transportation on motor vehicles. Finally, based on the IPAQ frequency and duration of the three types of PA assessed (walking, MPA and VPA), participants’ PA levels were classified in four categories following a stroke-adapted criteria (Table 1) [3].

#### 2.2.3. Physical Function Assessment

The TUG test [33] was used as a measure of functional mobility. Starting in a seated position, participants were asked to rise from a chair upon the evaluator’s command, walk a 3 m distance at a comfortable and safe pace, turn around in the preferred direction, walk back to the chair and sit down. Participants were timed (in seconds) from standing up until they were seated. Longer times indicated worse functional mobility. Participants wore their usual footwear and were allowed to use an assistive device, if needed. A practice trial was performed before the timed trial, which was the one used for statistical analyses. This test has shown excellent validity in chronic stroke [49].

To assess lower limb strength, the 5 × STS was used [34]. During the test, the time the participants required to quickly get up and sit down from a chair 5 times was recorded. Shorter times were indicative of better strength. In chronic stroke, the 5 × STS has shown excellent inter-rater reliability, as well as validity [50].

Ambulatory level was determined by comfortable gait speed assessed with the 10 MWT [35]. Participants were asked to walk along a 10-m corridor at their preferred walking speed. Time was measured while participants walked the intermediate 6 m, to avoid the effects of acceleration/deceleration in the measurement. Speed was calculated as the timed-distance (6 m) divided by the time needed to travel this distance (m/s). Two trials were performed and the average of the two repetitions was used in the statistical analyses. This test has shown excellent inter-rater reliability [51] and validity in subjects with chronic stroke [49,52].

The Stroke Impact Scale 16 (SIS-16) [36] was used to assess self-reported stroke severity. This scale measures the impact of physical functioning on disability after stroke. It consists of 16 items, which assess basic and instrumental activities of daily living (7 items), mobility (8 items), and hand function (1 item) on a 5-level Likert scale (1 = inability to complete the item; 5 = not difficult at all). A summary score was obtained and then transformed into a percentage by using the following equation: [(Mean item score − 1)/5 − 1] × 100. Higher scores indicated better functioning. This scale has demonstrated excellent internal consistency and adequate validity [53].

Disability was assessed with the MRS [37]. Interview-delivered, participants were asked about their activities of daily living, involving outdoors activities. In the MRS, the subject’s physical, mental performance and speech should be considered when choosing a single possible grade that ranges from 0 (no disability) to 5 (disability requiring constant care for all needs). Psychometric characteristics of the MRS in stroke include excellent inter-rater reliability and concurrent validity [54,55].

### 2.3. Sample Size Estimation

Prior sample size estimation was based upon the significant correlation between the IPAQ-LF and accelerometer total PA time (correlation coefficient of 0.4) found in healthy older adults [20,21]. With α set at 0.05 and β at 0.15, a minimum sample size of 53 subjects was required [56]. However, a 15% risk of data loss or participants’ noncompliance was included in the calculation, determining a total sample size of 61 subjects.

### 2.4. Statistical Analysis

The data were analyzed using Matlab (The MathWorks, Inc., Natick, MA, USA, version R2018b) and IBM SPSS Statistics v.26 (SPSS Inc., Chicago, IL, USA) software. To report the main characteristics of participants, descriptive analyses were performed on the clinical and demographic variables. Normality was assessed for all continuous data. Hence, results are presented as mean (standard deviation), median (25–75th percentile) or frequencies (percentage), as appropriate.

To study validity against accelerometry, differences between the IPAQ self-report PA and SB variables versus accelerometer measures were studied using the Wilcoxon-signed rank test. Spearman’s rank correlations coefficients between each method (subjective versus objective) were calculated for total PA time, time spent in MVPA and in SB. Correlation coefficients were interpret as small 0.1 to 0.3, medium 0.3 to 0.5 and large 0.5–1 [57]. In addition, the Bland-Altman (BA) mean-difference plots with 95% Limits of Agreement (LoA) and the percentage error were used to assess the agreement and degree of concordance between the two methods (IPAQ-LF versus accelerometry). The percentage error of the LoA was calculated (1.96 × SD of bias/accelerometer mean of the data), and a cut-off value of ±30% was used [58]. BA analyses were performed for total PA time, time spent in MVPA and in SB. Also, to detect if any bias between measurements were fixed or proportional, regression analysis was performed on the BA plots.

Validity against physical function was determined using PA data from the IPAQ-LF as correlated with functional mobility (TUG), lower limb strength (5 × STS), ambulatory level (10MMT), stroke severity (SIS-16), and disability (MRS).

Finally, to look into the IPAQ-LF sensitivity and specificity, the ability of the questionnaire to classify subjects according to their PA level was studied. In this case, the agreement between the PA classification by the IPAQ-LF and the accelerometer-derived compliance was studied by using the Cohen’s Kappa coefficient, the Spearman’s correlation coefficient and the McNeemar test. For this purpose, the IPAQ-LF categories (Table 1) were dichotomized and those categorized as “very active” or “active” were classified as being sufficiently active. Sensitivity was defined as the probability of the IPAQ-LF to correctly identify individuals not meeting the international guidelines for PA (non-compliant) [46], while specificity was determined as the probability that a subject classified as insufficiently active by the IPAQ-LF is truly a non-compliant.

In all cases, significance was determined at the level of *p* < 0.05.

## 3. Results

Of the 61 subjects recruited, 56 participants were included in the analyses, as four subjects did not have valid acceleration data and one could not fully complete the IPAQ-LF. Table 2 displays participants’ demographic and clinical characteristics. Included participants wore the accelerometer an average of 5.86 (1.07) valid days and a mean of 789 (126.03) min/day.

### 3.1. Validity against Accelerometry

Table 3 outlines comparisons and correlations for the PA and SB variables obtained by the two studied methods. Compared to accelerometry, in the IPAQ-LF participants overestimated the min/week of MVPA, although this difference was not statistically significant. However, the total PA time was significantly underreported when assessed with the IPAQ-LF. The median of the time spent in SB in the IPAQ-LF was similar as that measured by accelerometry. Regarding the correlation analysis, the time spent in total PA resulted in a significant and large correlation coefficient between both methods.

The agreement between methods for the time spent in PA and SB is depicted in the BA plots of Figure 2. A mean difference of 75.6 (305.0) min of MVPA per week between the IPAQ-LF and the accelerometer data was found, indicating that the time spent in MVPA assessed with the IPAQ-LF was over-estimated. The LoA were wide, with a difference between 673 and −522 min/week (Figure 2a) and the LoA percentage error was 512.2%. The BA plot suggested the presence of proportional bias and the regression analysis showed a positive linear trend (R^2^ = 0.22) between the differences of both methods and the mean min/week. That is, the between-methods difference of the time spent in MVPA increased when participants reported higher MVPA time.

Results for the total PA time showed an average difference between IPAQ-LF and the accelerometer data of −585.2 (683.5) min/week, pointing to under-estimation of the reported total PA time in the IPAQ-LF. The LoA were wide (between 754.4 and −1924.8 min/week, Figure 2b) and their percentage error resulted in 127.7%. In the BA plot, the distribution of points on the scatterplot suggested proportional bias and the regression analysis showed a negative linear trend (R^2^ = −0.35) between the two methods differences and their average. In other words, the difference between IPAQ-LF-derived and accelerometer-measured total PA time decreased as the mean of the two methods increased.

Regarding SB, the mean difference between the IPAQ-LF and the accelerometer data was −1.61 (169.2) min/day, although the LoA were wide (330.1 to −333.3 min/day, Figure 2c) and their percentage error was high (53.2%). In this case, our participants under-reported the time spent in SB when answering the IPAQ-LF. The BA plot indicated measurement bias and the regression analyses presented no significant linear trend (R^2^ = 0.00) between differences and mean values.

### 3.2. Validity against Physical Function

The correlation analyses performed to study the IPAQ-LF validity against physical function are shown in Table 4. Time spent in MVPA and total PA time were significantly associated with all physical function tests. The highest correlation coefficients were found for the total PA time, which were medium-large. Time in SB was only correlated with SIS-16 and MRS, resulting in small-medium correlation coefficients.

### 3.3. IPAQ-LF Sensitivity and Specificity

The classification of participants by both methods (dichotomized IPAQ-LF categories and accelerometer-derived compliance) is shown in Table 5. The Kappa statistics between the IPAQ-LF and the accelerometry was not statistically significant (κ = 0.103, *p* = 0.353), as was so the correlation analysis (ρ = 0.124, *p* = 0.362), though the McNemar test was statistically significant (*p* = 0.002). However, when classifying participants with the IPAQ-LF, a medium level of sensitivity (64.2%) and high specificity (70%) in detecting insufficiently active subjects was observed.

## 4. Discussion

As far as we are aware, this study is the first to evaluate the validity of the IPAQ-LF in people with chronic stroke. Results showed that, for total PA time, the correlation between the IPAQ-LF and accelerometry was statistically significant and large. In the IPAQ-LF, those participants with higher PA levels tended to over-report MVPA time and under-report total PA when compared to accelerometry. Both methods measured sedentary time similarly, though random error was observed between them. The IPAQ-LF showed an acceptable ability to correctly classify insufficiently active individuals. In addition, the association between the IPAQ-LF variables and physical function parameters resulted in moderate–strong correlations.

When validity was assessed through its association with accelerometry, our results are in agreement with previous validation studies of the IPAQ-LF performed in population with physical impairments [15,21,23]. Time spent in MVPA, as well as in SB, correlated weakly. However, for total PA time a large significant correlation was found (ρ = 0.55), which agrees with that considered acceptable between accelerometer data and self-report total PA (≥0.5) for a PA questionnaire [59]. This confirms the hypothesis of the present study.

In this study, the weak correlation observed for the time spent in MVPA between the IPAQ-LF and accelerometry could be explained by the way both methods measure walking activity. In the IPAQ-LF, walking is always considered MVPA. However, accelerometry depends on the acceleration of the body movement. In this sense, it is frequent that people post-stroke show low walking speed (<0.8 m/s), which consequently may be registered as LPA by the accelerometer since body acceleration may not meet the threshold for MVPA [60]. In this regard, subjects with low walking speed represented 44.6% of our sample. This is in line with the clinical reality of stroke individuals, where heterogeneity due to the extremely variable impairments and disabilities that occur after stroke is a major feature [24].

Over-reporting MVPA by the IPAQ-LF has been noticed previously in elderly and in diseased populations [17,20,21,23,24,25]. In our study, overestimating MVPA was found in those participants with higher levels of PA. Several reasons may account for this finding. First, due to physical impairments and deconditioning after stroke, performing certain activities of daily living may involve difficulty or effort. In our experience, these activities could have been considered MVPA as asked following the IPAQ-LF instructions. However, they should be categorized as LPA according to the Ainsworth Compendium of Physical Activity, in which the IPAQ is based. Commonly, people with stroke have proved objectively to mainly perform low intensity PA [61], which coincide with most of the activities of daily living. Hence, as it has been found in elderly subjects [20], participants of our study may have reported a relative amount of LPA as if it was MVPA. Second, duplicity of entries across domains for activities with the same level of intensity may have accumulated small overestimations for each separate PA domain, resulting in greater overestimation for total self-reported MVPA. Third, though misunderstanding was tried to be avoided by administering the IPAQ in an interview way, some of our participants presented a mild cognitive impairment (MoCA: MD = 23.5, IQR = 20.0–25.8), which may have influenced their comprehension and recall ability. However, the participants’ activity diaries served as a support for interviewers to help them when recalling PA, as well as SB.

On the other hand, our results revealed that total PA time was underestimated in our cohort, matching previous findings where the IPAQ-LF was used in elderly people [20,21]. While accelerometers record activities of all intensity (from very low to very vigorous), the IPAQ-LF is only focused on MVPA and does not take into account LPA. Thus, as mentioned by van Holle et al. [20], when addressing total PA, a way to improve the validity of the IPAQ-LF could be to include questions about LPA. This can also be applied in the chronic stroke population. Furthermore, in the IPAQ-LF, only those activities lasting 10 min or longer are recorded and those not achieving this minimum duration are ignored, while the accelerometer records PA all the time. This is quite relevant if we take into account that reported patterns of daily activities in post-stroke individuals are characterized by low frequency and duration [3]. In this regard, validity results may have been improved if accelerometry PA were quantified as activity bouts lasting ≥10 consecutive minutes. However, we followed the 2020 World Health Organization guidelines on PA and SB, which highlight the importance of total PA volume regardless of the length of the bout [62]. Consequently, future research about the development of a stroke-adapted version of the IPAQ-LF is needed.

The findings of the present study revealed moderate to strong correlations between the PA assessed by the IPAQ-LF and those physical parameters that generally impair physical function in subjects with chronic stroke, such as functional mobility, lower limb strength, ambulatory level, stroke severity and disability. Our results have proven similar validity when compared with other questionnaire in chronic stroke population (the Stroke Physical Activity Questionnaire), which also moderately correlated (ρ = 0.3–0.4) with the same physical function parameters [63]. Moreover, given that a correlation coefficient between PA questionnaires and physical function variables >0.30 has been reported to be acceptable, the IPAQ-LF exhibited moderate validity.

The ability of the IPAQ-LF to correctly classify insufficiently active individuals was acceptable. The sensitivity was medium and the specificity was high in our chronic stroke population. In this study, a classification criterion specifically designed for the stroke population was employed [3]. Hence, when using this classification criterion, there is a reasonably medium-high probability to correctly classify subjects in the appropriate PA level category. This could be of great interest when screening inactive subjects post-stroke when trying to implement clinical strategies to enhance healthy habits.

Validation studies of PA questionnaires to assess PA and SB in subjects with chronic stroke are scarce. As far as we know, there is only one other research work [63] where a new questionnaire, the Stroke Physical Activity Questionnaire, was developed and validated against the IPAQ shortform. However, comparisons with an objective measure as accelerometry had not been performed. Moreover, our results concur with those previously found when the IPAQ-LF was validated against accelerometry in elders [15,17,20,21,22] and disease-related populations [23,24,25,26], which also found acceptable correlation coefficients and low agreement.

One of the strengths of this study was the face-to-face interview administration of the IPAQ-LF. The guidance by trained interviewers proved beneficial in elderly population, obtaining higher accurate responses [26]. Since cognitive impairment may be present after stroke, administering the IPAQ by interview could prevent misinterpretation of common PA terms (e.g., duration, frequency, and intensity), facilitate participants recall activities that otherwise might have been forgotten and help interviewees in estimating duration of activities. It could also be useful in detecting possible overlap or duplicate over-reporting when referring to certain activities, such as walking for leisure and walking for transport. Another strength of this study was that participants fulfilled PA diaries during the monitoring period. The daily break down of typical activities registered in the PA diary made the reporting of activities and their duration estimation easier. Undoubtedly, this prevented recall bias since it was found difficult to accurately remember bouts of PA over the past week. Nevertheless, despite taking these careful measures, bias could have occurred since not all the participants fully completed the PA diary and social desirability could not be totally avoided.

It is plausible that a number of limitations could have influenced our results. In this study, gold standard measures of PA (e.g., doubly labelled water) were unfeasible to be implemented for measuring usual free-living PA over a 7-day period. However, accelerometers are often used for validating PA questionnaires since they can accurately discriminate between frequency and intensity of PA and are considered to be one of the most suitable instruments to objectively measure PA [15,17,20,23]. Similarly, as in all accelerometry studies, the data processing criteria (runs of zeroes, number of valid min/day, number of valid days/week and the cut-points) could have affected our outcomes. Following previous studies about actigraphy in stroke population and elders [27,39,43], we decided to use Freedson et al.’s [44] cut-points. Also, dismissing the truncation rules of high values in the IPAQ-LF may have influenced our results, as the IPAQ data was expected to be more similar to accelerometry data. However, our findings agree with those previously reported in elders [15,17,20,21,22] and disease-related populations [23,24,25,26] when truncation was used. Moreover, as clinicians do not usually truncate PA time when assessing it, dismissing truncation could more reliably reflect the use of this questionnaire in clinical practice and improve its applicability to this scope. Hence, as stated in the IPAQ scoring protocol, this rule requires further testing [47]. On the other hand, the generalizability of our results could be improved with a larger sample size. However, the number of participants (*n* = 56) was sufficient in accordance with an a priori power analysis. In addition, generalizability could also be affected by our participants’ sex, although our sample characteristics agree with the sociodemographic trends observed in the stroke population [64,65] and recent literature showed no significant differences in PA levels between men and women [66]. In that sense, it should be noted that, due to the heterogeneity of this population, further studies analyzing the validity of the IPAQ-LF by subgroups (e.g., age, sex, type of stroke, walking speed, cognitive impairment …) are needed. Furthermore, future research with IPAQ-LF could address differences by age, sex, and type of stroke in PA levels within its four domains (job-related, transport-related, domestic, and leisure-time PA). This knowledge could help to identify which domains concentrate less PA levels by subgroups and to develop strategies to reduce sedentary behavior after stroke [67].

Identifying PA and SB patterns in people with chronic stroke is essential in order to tailor PA interventions and maximize health benefits. For this purpose, PA questionnaires validated in this population are relevant since they are a simple, inexpensive, and efficient way to assess PA and SB in clinical settings. Consequently, we believe our paper makes a valuable contribution to the knowledge of this field.

## 5. Conclusions

According to the results of the present study, it may be concluded that, in people with chronic stroke, the IPAQ-LF (interviewed-base, 7-day recall) presents acceptable levels of validity for estimating total PA time in those subjects insufficiently active. Consequently, with the IPAQ-LF, clinicians can detect individuals with chronic stroke not accumulating sufficient levels of PA. However, the IPAQ-LF overestimates MVPA and underestimates total PA in those more physically active, since it does not consider LPA, which is the most usual PA in stroke survivors. Hence, as an assessment measure of PA and SB, and taking into account our results, the IPAQ-LF could be a valid tool to screen for inactive individuals with chronic stroke who can benefit from PA interventions addressed to implement healthier lifestyles.

## Figures and Tables

**Figure 1 ijerph-18-04729-f001:**
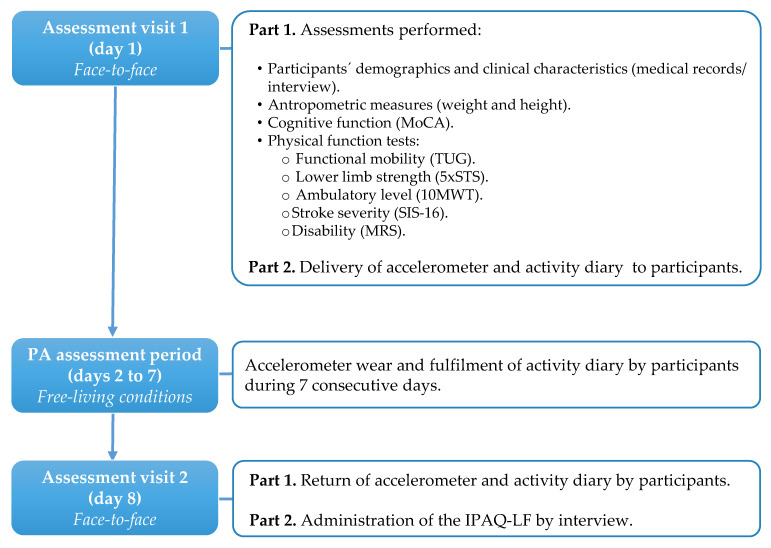
Flowchart of the study design. PA: physical activity; MOCA: Montreal Cognitive Assessment; TUG: timed up and go test; 5 × STS: five times sit-to-stand test; 10 MWT: ten-meter walk test; SIS-16: Stroke Impact Scale 16; MRS: Modified Rankin Scale; IPAQ-LF: International Physical Activity Questionnaire Long Form.

**Figure 2 ijerph-18-04729-f002:**
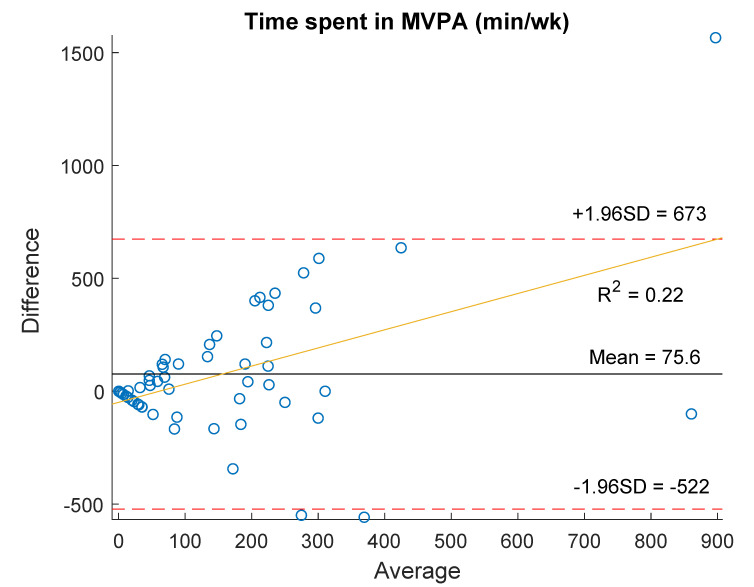
Bland-Altman plots showing the difference vs average between the IPAQ-LF and accelerometry in: (**a**) time spent in moderate-to-vigorous intensity physical activity; (**b**) total physical activity time; (**c**) time spent in sedentary behavior. Solid black lines represent the mean of the differences, dotted red lines indicate the 95% limits of agreement and solid yellow lines represents the linear regression model.

**Table 1 ijerph-18-04729-t001:** Criteria used to categorize participants according to their self-reported physical activity [3].

Category	Criteria *
Very active	Option 1: ≥5 days/week and ≥30 min/session of VPA OR Option 2: ≥3 days/week and ≥20 min/session of VPA + ≥5 days/week and ≥30 min/session of MPA or walking
Active	Option 1: ≥3 days/week and ≥20 min/session of VPA OR Option 2: ≥5 days/week and ≥30 min/session of MPA or walking OR Option 3: ≥5 days/week and ≥ 150 min/week of any combination (VPA + MPA+ walking)
Irregularly active: This category is divided into two subgroups:	
Irregularly active a	≥5 days/week OR ≥150 min/week of any combination (VPA + MPA+ walking)
Irregularly active b	Not meeting any of the recommendation criteria (frequency OR duration) of type A
Sedentary	Not engage in at least 10 min of continuous activity during the week

* To be classified in one physical activity level, participants have to meet one of the corresponding options. MPA: moderate-intensity physical activity; VPA: vigorous-intensity physical activity; MVPA: moderate-to-vigorous intensity physical activity.

**Table 2 ijerph-18-04729-t002:** Participants’ demographic and clinical characteristics (*n* = 56).

Variable	Value
Sex, male/female, *n* (%)	37 (66.1)/19 (33.9)
Age (years)	58.1 (11.1)
Body mass index (kg/m^2^)	28.7 (4.6)
Type of stroke: ischemic/hemorrhagic, *n* (%)	34 (60.7)/22 (39.3)
Side of hemiparesis: Left/Right, *n* (%)	33 (58.9)/23 (41.1)
Time since stroke (months), median (IQR)	63.0 (37.8–105.8)
MOCA, median (IQR)	23.5 (20.0–25.8)
FACHS, *n* (%)	
(i) 2	10 (17.8)
(ii) 3	12 (21.4)
(iii) 4	27 (48.2)
(iv) 5	7 (12.5)
Gait aid, *n* (%)	
(i) Single cane	19 (33.9)
(ii) Tripod cane	3 (5.4)
(iii) Walker	1 (1.8)
(iv) Ankle-foot orthosis	18 (32.1)
TUG test (seconds), median (IQR)	13.2 (10.7–20.8)
5 × STS (seconds), median (IQR)	14.6 (12.2–17.7)
10 MWT, comfortable walking speed, (m/s)	0.8 (0.4)
SIS-16, median (IQR)	86.7 (18.36)
MRS, *n* (%)	
(i) 0	2 (3.6)
(ii) 1	9 (16.1)
(iii) 2	24 (42.9)
(iv) 3	14 (25.0)
(v) 4	7 (12.5)

Values are mean (standard deviation) unless otherwise indicated. IQR: interquartile range (25–75th percentile); MOCA: Montreal Cognitive Assessment; FACHS: Functional Ambulation Classification of Hospital of Sagunto; TUG: Timed up and go test; 5 × STS: five times sit-to-stand test; 10 MWT: ten-meter walk test; SIS-16: Stroke Impact Scale 16; and MRS: Modified Rankin Scale.

**Table 3 ijerph-18-04729-t003:** Comparison and correlation analysis of physical activity and sedentary behavior data as measured by the IPAQ long-form and the accelerometer.

Variable	IPAQ Long-Form	Accelerometer	Wilcoxon Z (*p*-Value)	Spearman’s Rho (*p*-Value)
Time spent in MVPA (min/wk)	95.0 (0.0–265.0)	43.4 (19.3–162.4)	−1.39 (0.16)	0.10 (0.45)
Total PA time (min/wk)	372.5 (183.8–736.5)	940.8 (560.4–1421.5)	−5.44 (<0.001) *	0.55 (<0.001) *
Time spent in SB (min/d)	617.1 (565.9–705.7)	621.5 (553.1–729.9)	−0.03 (0.97)	0.04 (0.79)

Results are expressed as median interquartile range (25–75th percentile). * Indicates significant differences. IPAQ: International Physical Activity Questionnaire; MVPA moderate-to-vigorous intensity physical activity; PA: physical activity; SB: sedentary behavior.

**Table 4 ijerph-18-04729-t004:** Association between the IPAQ long-form and physical function.

IPAQ Long-Form Data	TUG Test (s)	5 × STS (s)	10 MWT (m/s)	SIS-16	MRS
Time spent in MVPA (min/wk)	−0.38; 0.004 *	−0.29; 0.047 *	0.41; 0.002 *	0.34; 0.011 *	−0.47; <0.001 *
Total PA time (min/wk)	−0.52; <0.001 *	−0.31; 0.030 *	0.47; <0.001 *	0.38; 0.004 *	−0.60; <0.001 *
Sedentary time (min/d)	0.23; 0.09	0.12; 0.43	−0.23; 0.09	−0.29; 0.029 *	0.31; 0.021 *

Results are shown as Spearman’s rho; *p*-Value. * Indicates significant differences. IPAQ: International Physical Activity Questionnaire; TUG: Timed up and go test; 5 × STS: five times sit-to-stand test; 10 MWT: ten meter walk test; SIS-16: Stroke Impact Scale 16 and MRS: Modified Rankin Scale; MVPA moderate-to-vigorous intensity physical activity; PA: physical activity; SB: sedentary behavior.

**Table 5 ijerph-18-04729-t005:** Distribution of participants according to their physical activity level categorized by the IPAQ-LF and by accelerometry.

		Accelerometer Category		Total
		Compliant	Non-Compliant	
IPAQ-LF category	Sufficiently active	21	5	26
	Insufficiently active	21	9	30
Total		42	14	56

IPAQ-LF: International Physical Activity Questionnaire long-form.

## Data Availability

The data underlying this article will be shared on reasonable request to the corresponding author.

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
