# Peer review of "Validity of the International Physical Activity Questionnaire Long Form for Assessing Physical Activity and Sedentary Behavior in Subjects with Chronic Stroke"

_ijerph, 2021, doi:10.3390/ijerph18094729_

Round 1

Reviewer 1 Report

This paper demonstrations the validity of IPAQ for assessing physical activity and sedentary behavior. The work of this paper is practical and logical. However, there are some problems to be further improved as well: The chart in this paper should show the content more clearly, such as,

significant differences in Table 4 are not obvious, the author can try to use (*) to indicate.

The experimental process should be written more clearly.

Lack of some detailed descriptions, such as, the specific division of MVPA and VPA.

There are still some writing problems in the article, such as missing punctuation, sentence grammar, please check carefully before submitting your paper, so that your paper will be more perfect.

Author Response

We thank the reviewer 1 for his/her critical review and valuable comments. We have taken into account all of his/her recommendations and suggestions. Itemized responses are listed below. All the modifications have been clearly highlighted using the "Track Changes" throughout the manuscript to make its revision easier (See ijerph-1195285 marked up).

Reviewer 2 Report

Maria-Arantzazu Ruescas-Nicolau and coll.   analysed the correlation between the use of Accellerometer (Actigraph3GTX+) and the  IPAQ-LF questionnaire in order to identify the physical activity time spent in Low- Moderate or Sedentary in a cross-sectional study  in community living 16 adults with post-stroke sequelae (≥6 months) and preserved ambulation. 

The study result original, well designed; the methods are well depicted; statistical analyses was appropriated and strong; the conclusions are in line to the aims.

Minor revisions:

The authors  must discuss in greater details the influence of sex and type of stroke on the PA levels in particular on the results obtained from Actigraph3GTX and in comparison to IPAQ-LF questionnaire.

Author Response

We thank the reviewer 2 for his/her critical review and valuable comments. We have taken into account all of his/her recommendations and suggestions. Itemized responses are listed below. All the modifications have been clearly highlighted using the "Track Changes" throughout the manuscript to make its revision easier (See ijerph-1195285 marked up).

Reviewer 3 Report

This paper describes the Validity of the International Physical Activity Questionnaire Long Form for assessing physical activity and sedentary behavior in subjects with chronic stroke, using accelerometer data as reference in a reasonable large cohort of enrolled post-stroke patients.

The paper is well written and clearly understandable.

MAIN COMMENTS

Line 171: “To create a MVPA variable” à can you rephrase better? It is not clear.

Lines 162-179: this paragraph is very important since the applied methodology impacts results. I found some explanations not always clearly understandable as the rest of the paper is. I suggest revising for more clarity.

“The intention was to maintain the participants´ original responses since, if guidelines were applied, answers would have changed, which was considered inappropriate.” à please explain better why applying the guidelines modifies the data in an uncoherent way. Or, cite some (more) studies that already followed your approach.

MINOR

Since many clinical and accelerometric measures are performed, you may consider to introduce a figure to show the study design: measurements, when they are performed, and what you compare (suggested, not requested).

Abstract: “The 18 ActiGraph GT3X+” please, mention what it is for clarity. Es: …the ActiGraph accelerometer, …

BAPlots: ylabel: I would suggest removing “Difference between” (which is quite ugly to read). Consider increasing quality of figures for final drafting (e.g.: Matlab “print” command)

DISCUSSION

Can you comment more the effects induced in results with the modifications of the guidelines proposed in this study?

Author Response

We thank the reviewer 3 for his/her critical review and valuable comments. We have taken into account all of his/her recommendations and suggestions. Itemized responses are listed below. All the modifications have been clearly highlighted using the "Track Changes" throughout the manuscript to make its revision easier (See ijerph-1195285 marked up).
